# Study on the Staged Operation of a Multi-Purpose Reservoir in Flood Season and Its Effect Evaluation

Chongxun Mo [1,2,3], Can Zhu [1,2,3], Yuli Ruan [1,2,3,]*, Xingbi Lei [1,2,3], Zhenxiang Xing [4] and Guikai Sun [1,2,3]

1 College of Architecture and Civil Engineering, Guangxi University, Nanning 530004, China; mo_209@163.com (C.M.); zhucan12580@163.com (C.Z.); leixingbi@foxmail.com (X.L.); gksungxu@163.com (G.S.)
2 Key Laboratory of Disaster Prevention and Structural Safety of the Ministry of Education, Nanning 530004, China
3 Guangxi Key Laboratories of Disaster Prevention and Engineering Safety, Nanning 530004, China
4 School of Water Conservancy and Civil Engineering, Northeast Agricultural University, Harbin 150006, China; zxxing@neau.edu.cn
* Correspondence: yuliruan777@163.com

**Abstract:** A reasonable analysis of flood season staging is significant to the management of floods and the alleviation of water shortage. For this paper, the case of the Chengbi River Reservoir in China was selected for study. Based on fractal theory, the flood season is divided into several sub-seasons by using four indexes (multi-year average daily rainfall, multi-year maximum rainfall, multi-year average daily runoff, and multi-year maximum daily runoff) in this study. Also the Benefit-Risk theory is applied to evaluate the effects of staged dispatching. The results show that the flood season of the Chengbi River basin should be divided into the pre-flood season (13 April–6 June), the main flood season (7 June–9 September) and the post-flood season (10 September–31 October). After adjusting the flood limit water level for sub-season and benefit assessment, the probability of exceedance after reservoir flood season operation increases by $0.13 \times 10^{-5}$, the average annual expected risk is 0.2264 million RMB, and the average annual benefit increases by 0.88–1.62 million RMB. The benefits obtained far outweigh the risks, indicating the importance of staging the flood season.

**Keywords:** effect evaluation; fractal method; flood season staging; multi-purpose reservoir; probability of exceedance; staged operation





## 1. Introduction

According to the statistics of the United Nations Environment Program, compared to the past 100 years, the global annual water resources per capita have reduced from 40,000 $m^3$ to 6840 $m^3$. Additionally, it is expected that by 2030, nearly 50% of the world's population will have less than 1000 $m^3$ of annual per capita water resources and will be in a state of severe water shortage [1]. With the development of society and growing populations, water shortages are becoming more and more prominent. However, large amounts of water have been discharged during the flood season, resulting in a huge waste of water resources. Nowadays, the use of floods has become more and more important in most areas [2]. Reservoir scheduling is an effective way to utilize flood resources and has been studied by a large number of scholars [3–6]. In most of the countries, floods have seasonal patterns of change, and it is necessary to study the flood season and stage it rationally to raise the FLWL (flood limit water level) of the reservoir appropriately. In this way, flood resources can be used to a greater degree, which is one of the important issues that needs to be studied and solved today.

There have been many studies on the seasonal patterns and staging of floods during the flood season [7–12]. Different staging methods have been used to segment the flood season, such as the probabilistic change-point analysis technique [13], the vector statistic and relative frequency method [14–17], and the fuzzy set method [18–21]. As for the

selection of indicator factors for staging, most previous studies have used a single factor to stage the flood season. For example, peak flow [22] or average daily maximum flow [23] are used as a single indicator factor for flood staging studies, causing the staging results to not be mutually verified. A reasonable determination of the FLWL is the key to coordinate flood risk and reservoir benefit [24]. Therefore, many scholars have carried out extensive research on the optimization of FLWL. An FLWL model dynamic control was applied to reservoirs with indeterminate flood process lines, which effectively improved hydropower generation and flood utilization [25]. Liu et al. [26] optimized the design of staged flood limit levels, and a framework for optimal reservoir scheduling based on flood staging results was proposed [27].

In summary, most previous studies have used a single indicator factor for flood staging leading to uncertainty in the results. In addition, there is less involvement in the calculation of FLWLs for each phase and the evaluation of the benefits of the flood staging. Therefore, the objective of this study is to stage the flood season by selecting multiple indicator factors and then evaluate the benefits of the staging results. The Chengbi River reservoir is selected as the object of this study. Multi-year average daily rainfall time series, multi-year maximum rainfall time series, multi-year average daily runoff time series, and multi-year maximum daily runoff time series are used as index factors to divide flood season by fractal method. The benefit-risk theory is applied to evaluate the effects of staged dispatching.

## 2. Methodology

### 2.1. Fractal Method

According to fractal theory, hydrological processes that exhibit periodic changes over a certain period of time (influenced by deterministic factors) can be considered to be self-similar [28]. The occurrence of seasonality and timing of floods can be considered to have similar mechanisms, and so the fractal theory has been used in flood staging [29]. The fractal feature is described by capacity dimension. Assuming $F$ is a bounded subset of the d-dimensional Euclid space and $N(\varepsilon)$ is the least number of closures covering $F$ of radius $\varepsilon$, then the capacity dimension $D_b$ is defined as follows [30].

$$D_b = \lim_{\varepsilon \to 0}(\log N(\varepsilon) / \log(1/\varepsilon)) \tag{1}$$

The capacity dimension is calculated as follows:

(1) Take the sample point series $x_1, x_2, \cdots, x_n$ in the flood season, and determine the period length $T$ according to the start length and step span of the sample period, then select the flood season segmentation level $Y$ in period $T$ to reflect its sample.

(2) Take the time scale $\varepsilon = \{1d, 2d, \ldots, 10d\}$ and count the number of periods $N(\varepsilon)$ in which the sample $x_i$ exceeds the segmentation level $Y$. Calculate the corresponding relative time scales $NT(\varepsilon)$ and relative measures $NN(\varepsilon)$ from Equation (2) and Equation (3) based on $T$ and $\varepsilon$, and a linear fit to $\ln(\varepsilon)$ and $\ln NN(\varepsilon)$ to find the slope $b$ of the correlation. The capacity dimension $D_b$ is obtained from Equation (4).

$$NT(\varepsilon) = T/\varepsilon \tag{2}$$

$$NN(\varepsilon) = N(\varepsilon)/NT(\varepsilon) \tag{3}$$

$$D_b = 2 - b \tag{4}$$

(3) Change the period length $T$ and repeat the above steps. If the $D_b$ obtained is basically equal, then $T$ at this time is the same stage.

*2.2. Risk and Benefit Analysis Methodology*

2.2.1. Probability of Exceedance

Considering only the effect of flooding factors, the reservoir staging dispatch probability of exceedance calculation model is as follows.

$$P = P(q \geq Q) \tag{5}$$

where $q$ represents a random variable. *P-III* (Pearson type III) curve is generally used in flood peak discharge frequency curve. Then its density function can be expressed as

$$f(q) = \frac{\beta^{\alpha}}{\Gamma(\alpha)}(q - a_0)^{\alpha-1}e^{-\beta(q-a_0)} \tag{6}$$

$\Gamma(\alpha)$—The gamma function of $\alpha$.

$\alpha, \beta, a_0$—Three parameters characterizing the shape, scale and location of *P-III* distribution. $\alpha > 0, \beta > 0$.

The cumulative distribution function can be expressed as

$$P = P(q \geq Q) = \int_{Q}^{\infty} f(q)\mathrm{d}q \tag{7}$$

2.2.2. Benefit Analysis

In the benefit analysis, the reservoir capacity should be calculated according to the actual situation. The increased capacity will not only bring direct benefits in terms of power generation and water supply, but also generate indirect economic benefits such as irrigation, farming, tourism, etc. Water supply and power generation benefits are calculated using the following formula.

$$W = V \times \eta \tag{8}$$

where $W$ is the economic benefit from water supply or electricity generation, $V$ is the additional storage capacity after adjusting the FLWL, and $\eta$ is the economic efficiency for one cubic meter of water.

## 3. Study Area and Data

The Chengbi River Reservoir is located in Baise City, Guangxi Province, downstream of Chengbi River, (106°21′ E–106°48′ E, and 23°50′ N–24°45′ N) (Figure 1). It is the second-largest earth-rock fill dam reservoir project in China, with a total storage capacity of 1.15 billion m³ and a normal storage level of 185 m. The engineering characteristics parameters of the reservoir are shown in Table 1. It operates under the rule of a single FLWL for the entire flood season, resulting in a low storage rate after floods and a large waste of flood resources. The average precipitation of the watershed over the years is 1560 mm, and the rainfall is unevenly distributed during the year, mostly concentrated in April to September, accounting for more than 85% of the annual rainfall. The flood season of the Chengbi River is from 13 April to 31 October with a low storage rate after flood season. The data selected in this paper are the daily precipitation and daily measured runoff from Ba Shou Station (BSS) from 1963 to 2016, and the four index factors (average daily rainfall, maximum daily rainfall, average daily runoff, and maximum daily runoff) which reflect the characteristics of the flooding period are used as the basic data of staging.

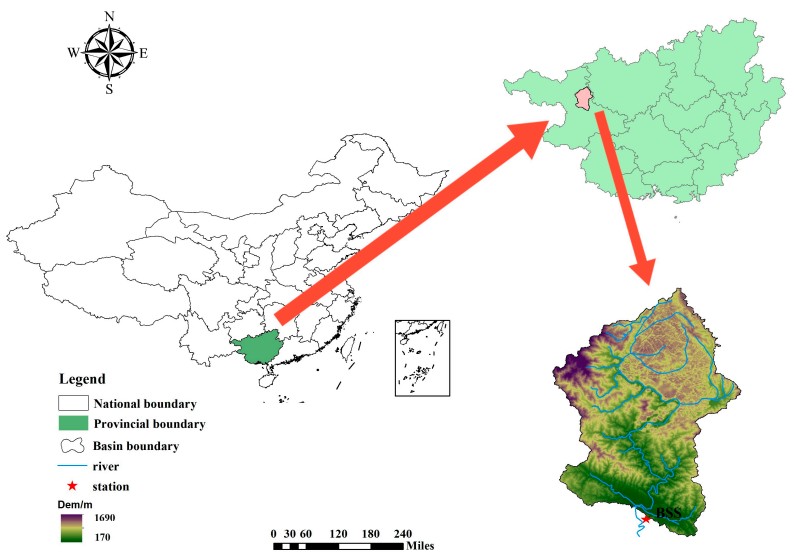

**Figure 1.** Location of the Chengbi River Reservoir.

**Table 1.** Engineering characteristics of the Chengbi River Reservoir.

| Name | Engineering Features | Value | Name | Engineering Features | Value |
|---|---|---|---|---|---|
| | water level (P = 0.1%) | 188.17 m | | summer mean temperature | 28.7 °C |
| | water level (P = 0.01%) | 189.30 m | | multi-year average runoff | 37.8 m$^3$/s |
| | water surface area | 39.1 km$^2$ | Hydrometeor | flood peak flow (P = 0.1%) | 6460 m$^3$/s |
| | dead water level | 167.00 m | | flood peak flow (P = 0.01%) | 7980 m$^3$/s |
| Reservoir and Dam | total storage | 1150 hm$^3$ | | crest level of the spillway | 176.00 m |
| | dead storage | 380 hm$^3$ | | crest level of the gates | 185.70 m |
| | crest level of dam | 190.40 m | Spillway | Width of spillway | 4 × 12 m |
| | height of dam | 70.40 m | | flood discharge capacity (P = 0.1%) | 3040 m$^3$/s |
| | crest length | 425.00 m | | flood discharge capacity (P = 0.01%) | 3580 m$^3$/s |

## 4. Results

### 4.1. Flood Staging Results

The fractal calculation was carried out based on the runoff and rainfall data of the Chengbi River Reservoir from 1963 to 2016. When calculating the capacity dimension $D_b$, taking into account the seasonal characteristics of the flood change pattern and its causes, the staging is generally not shorter than 30 days [31]. According to scholars [32,33], the maximum deviation of the capacity dimension of a fractal is less than 5% classified as a class. The initial length of the study is 30 d, then 10 d as a step to calculate the $D_b$, and finally shortened to 5 d for the calculation. The results are shown in Tables 2–5.

From Table 2, it can be seen that the maximum deviation of the $D_b$ at $T = 30$ d $\sim 55$ d is 1.96% (<5%) of the minimum capacity dimension, so it is in the same stage. When $T = 60$ d, it is not considered to be in the same stage as the previous time period, because the value of $D_b$ is mutated and the maximum deviation is 13.05% (>5%). Therefore, the pre-flood season can be identified as 13 April to 6 June. A sudden change in the value of the $D_b$ at time period $T = 85$ d, with a maximum relative error of 1.74% in the preceding time period, which can be classified as the second stage. Accordingly, the main flood season can be identified as 7 June to 25 August, and the duration of the post-flood season is from 26 August to 31 October.

Using a relative error equal to 5% as the threshold for whether or not it is the same stage, it is clear from Tables 3 and 4 that the flood season can be divided into three phases. The flood segmentation results by using average daily maximum rainfall as an indicator factor is as follows: the pre-flood season (13 April to 6 June), the main flood season (7 June

to 30 August), the post-flood season (31 August to 31 October). The flood segmentation results by using average daily maximum rainfall as an index factor is as follows: the pre-flood season (13 April to 11 June), the main flood season (12 June to 30 August), and the post-flood season (31 August to 31 October). As shown in Table 5, the flood segmentation using multi-year average daily maximum runoff can be divided into four phases, but according to the Code for Hydraulic Calculations for Water Projects, a flood should not be divided into more than three sub-flood seasons. Therefore, based on the multi-year runoff characteristics of the Chengbi River basin, combining the first and second phases into one as the pre-flood season. Then, the pre-flood season is from 13 April to 16 June, the main flood season is from 17 June to 9 September, and the post-flood season is from 10 September to 31 October.

**Table 2.** Staging results of multi-year average daily rainfall.

| Stages | Duration $T$ (d) | Threshold Levels (mm) | Staged Period | Slope b | $D_b$ | Relative Error (%) |
|---|---|---|---|---|---|---|
| The first stage | 30 | 65 | 13 April–12 May | 0.547 | 1.453 | — |
| | 40 | 71 | 13 April–22 May | 0.575 | 1.425 | 1.96 |
| | 50 | 75 | 13 April–1 June | 0.569 | 1.431 | 1.96 |
| | 55 | 74 | 13 April–6 June | 0.563 | 1.437 | 1.96 |
| | 60 | 75 | 13 April–11 June | 0.389 | 1.611 | 13.05 |
| The second stage | 30 | 113 | 7 June–6 July | 0.499 | 1.501 | — |
| | 40 | 105 | 7 June–16 July | 0.493 | 1.507 | 0.40 |
| | 50 | 100 | 7 June–26 July | 0.509 | 1.491 | 1.07 |
| | 60 | 101 | 7 June–5 August | 0.49 | 1.51 | 1.27 |
| | 70 | 104 | 7 June–15 August | 0.483 | 1.517 | 1.74 |
| | 80 | 98 | 7 June–25 August | 0.489 | 1.511 | 1.74 |
| | 85 | 97 | 7 June–30 August | 0.342 | 1.658 | 11.2 |
| The third stage | 30 | 65 | 26 August–24 September | 0.529 | 1.471 | — |
| | 40 | 64 | 26 August–4 October | 0.519 | 1.481 | 0.68 |
| | 50 | 65 | 26 August–14 October | 0.523 | 1.477 | 0.68 |
| | 60 | 62 | 26 August–24 October | 0.523 | 1.477 | 0.68 |
| | 67 | 62 | 26 August–31 October | 0.501 | 1.499 | 0.68 |

**Table 3.** Staging results of multi-year maximum daily rainfall.

| Stages | Duration $T$ (d) | Threshold Levels (mm) | Staged Period | Slope b | $D_b$ | Relative Error (%) |
|---|---|---|---|---|---|---|
| The first stage | 30 | 65 | 13 April–12 May | 0.568 | 1.432 | — |
| | 40 | 71 | 13 April–22 May | 0.575 | 1.425 | 0.49 |
| | 50 | 75 | 13 April–1 June | 0.568 | 1.432 | 0.49 |
| | 55 | 74 | 13 April–6 June | 0.562 | 1.438 | 0.91 |
| | 60 | 75 | 13 April–11 June | 0.489 | 1.511 | 6.04 |
| The second stage | 30 | 113 | 7 June–6 July | 0.580 | 1.42 | — |
| | 40 | 104 | 7 June–16 July | 0.581 | 1.419 | 0.85 |
| | 50 | 100 | 7 June–26 July | 0.569 | 1.431 | 0.85 |
| | 60 | 100 | 7 June–5 August | 0.561 | 1.439 | 1.41 |
| | 70 | 103 | 7 June–15 August | 0.585 | 1.415 | 1.70 |
| | 80 | 98 | 7 June–25 August | 0.585 | 1.415 | 1.70 |
| | 85 | 97 | 7 June–30 August | 0.571 | 1.429 | 1.70 |
| | 90 | 96 | 7 June–4 September | 0.469 | 1.531 | 8.20 |
| The third stage | 30 | 61 | 31 August–29 September | 0.423 | 1.577 | — |
| | 40 | 58 | 31 August–9 October | 0.461 | 1.539 | 2.47 |
| | 50 | 62 | 31 August–19 October | 0.453 | 1.547 | 2.47 |
| | 62 | 60 | 31 August–31 October | 0.473 | 1.527 | 3.27 |

**Table 4.** Staging results of multi-year average daily runoff.

| Stages | Duration $T$ (d) | Threshold Levels (m³/s) | Staged Period | Slope b | $D_b$ | Relative Error (%) |
|---|---|---|---|---|---|---|
| The first stage | 30 | 10 | 13 April–12 May | 0.185 | 1.815 | — |
| | 40 | 15 | 13 April–22 May | 0.183 | 1.817 | 0.11 |
| | 50 | 24 | 13 April–1 June | 0.207 | 1.793 | 1.34 |
| | 60 | 32 | 13 April–11 June | 0.184 | 1.816 | 1.34 |
| | 65 | 36 | 13 April–16 June | 0.115 | 1.885 | 5.13 |
| | 70 | 40 | 13 April–21 June | 0.047 | 1.953 | 8.92 |
| The second stage | 30 | 105 | 12 June–11 July | 0.568 | 1.432 | — |
| | 40 | 106 | 12 June–21 July | 0.581 | 1.419 | 0.92 |
| | 50 | 109 | 12 June–31 July | 0.540 | 1.460 | 2.89 |
| | 60 | 110 | 12 June–10 August | 0.585 | 1.415 | 3.18 |
| | 70 | 111 | 12 June–20 August | 0.557 | 1.443 | 3.18 |
| | 80 | 108 | 12 June–30 August | 0.568 | 1.432 | 3.18 |
| | 85 | 107 | 12 June–4 September | 0.449 | 1.551 | 9.61 |
| The third stage | 30 | 53 | 31 August–29 September | 0.097 | 1.903 | — |
| | 40 | 47 | 31 August–9 October | 0.096 | 1.904 | 0.05 |
| | 50 | 43 | 31 August–19 October | 0.110 | 1.890 | 0.75 |
| | 62 | 39 | 31 August–31 October | 0.132 | 1.868 | 1.93 |

**Table 5.** Staging results of multi-year maximum daily runoff.

| Stages | Duration $T$ (d) | Threshold Levels (m³/s) | Staged Period | Slope b | $D_b$ | Relative Error (%) |
|---|---|---|---|---|---|---|
| The first stage | 30 | 60 | 13 April–12 May | 0.434 | 1.566 | — |
| | 35 | 73 | 13 April–17 May | 0.432 | 1.568 | 0.13 |
| | 40 | 84 | 13 April–22 May | 0.267 | 1.733 | 10.66 |
| The second stage | 30 | 216 | 18 May–16 June | 0.573 | 1.427 | — |
| | 35 | 258 | 18 May–21 June | 0.492 | 1.508 | 5.67 |
| | 40 | 238 | 18 May–26 June | 0.435 | 1.565 | 9.67 |
| The third stage | 30 | 318 | 17 June–16 July | 0.523 | 1.477 | — |
| | 40 | 321 | 17 June–26 July | 0.528 | 1.472 | 0.34 |
| | 50 | 307 | 17 June–5 August | 0.506 | 1.494 | 1.49 |
| | 60 | 307 | 17 June–15 August | 0.502 | 1.498 | 1.77 |
| | 70 | 294 | 17 June–25 August | 0.537 | 1.463 | 2.39 |
| | 80 | 284 | 17 June–4 September | 0.504 | 1.496 | 2.39 |
| | 85 | 280 | 17 June–9 September | 0.490 | 1.510 | 3.20 |
| | 90 | 275 | 17 June–14 September | 0.419 | 1.581 | 8.07 |
| The fourth stage | 30 | 164 | 10 September–9 October | 0.385 | 1.615 | — |
| | 40 | 164 | 10 September–19 October | 0.423 | 1.577 | 2.41 |
| | 52 | 159 | 10 September–31 October | 0.417 | 1.583 | 2.41 |

The results of the above calculations are summarized in the Table 6. Taking into consideration and based on the principle of extending the main flood season as much as possible, the results of the phasing were revised as follows: the pre-flood season is from 13 April to 6 June, the main flood season is from 7 June to 9 September, and the post-flood season is from 10 September to 31 October.

**Table 6.** Flood staging results for the Chengbi River Reservoir.

| Indicator Factors | The Pre-Flood Season | The Main Flood Season | The Post-Flood Season |
|---|---|---|---|
| Average daily rainfall | 13 April–6 June | 7 June–25 August | 26 August–31 October |
| Maximum rainfall | 13 April–6 June | 7 June–30 August | 31 August–31 October |
| Average daily runoff | 13 April–11 June | 12 June–30 August | 31 August–31 October |
| Maximum daily runoff | 13 April–16 June | 17 June–9 September | 10 September–31 October |

*4.2. Results of Flood Diversion Calculation*

4.2.1. Control Water Level

The maximum water level obtained from the flood regulation calculation in the main flood season is used as the control water level to raise the FLWL in other stages. Flood regulation calculations for the 1000-year and 10,000-year flood process lines in the main flood season starting at 185 m. The reservoir operation policy of flood regulation is that when the reservoir inflow is less than the rated storage outflow at the FLWL (1800 $m^3$/s), gates are used to control the outflow is equal to the incoming flow so that the water level in the reservoir could maintain the FLWL. When the reservoir inflow is greater than 1800 $m^3$/s, the gates are fully open to discharge flow. When the reservoir water level falls back to the FLWL, the discharge is controlled by the gates to keep the water level unchanged. The change of water level in the flood regulation calculation is shown in Figure 2. Then, the highest water level in the 1000-year flood regulation calculation in the Chengbi River Reservoir during the main flood season is 187.85 m, and the highest water level in the 10,000-year flood is 189.13 m, which are used as the control levels in the flood regulation calculation during the pre-flood season and post-flood season.

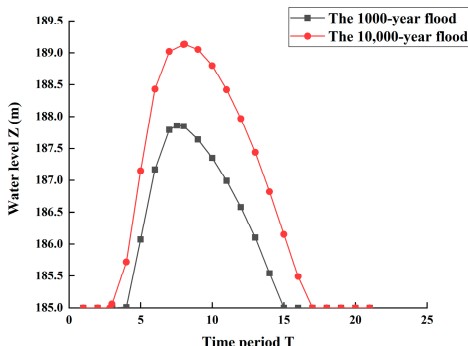

**Figure 2.** Variation of water level in flood regulation calculation of characteristic floods.

4.2.2. Flood Limit Water Level

Since raising the FLWL during the main flood season is not considered, only the different starting levels for the pre-flood season and the post-flood season are trialed. The initial water level is adjusted from 185 m and trialed in steps of 0.5 m. The results are summarized in Table 7. It can be seen that the maximum water level will not exceed the control level of 187.85 m when the FLWL is between 185.0 and 187.5 during the flood regulating calculation at the 1000-year flood process line in the pre-flood season. In the 10,000-year flood process line flood adjustment calculation, when the FLWL is 185.0 to 188.0 m, the maximum water level will not exceed the control level of 189.13 m. Therefore, the FLWL of the pre-flood season is set between 185.0 and 187.5 m without reducing the flood control standard of the reservoir. Correspondingly, the FLWL in the post-flood season is set between 185.0 and 187.5 m.

To sum up, the FLWL of Chengbi River reservoir in the pre-flood season and the post-flood season can be increased to some extent. However, for the pre-flood season in April and May, it makes little sense to raise the FLWL. Firstly, it is the time when agriculture in Baise City needs a lot of water, so it becomes impractical to raise the FLWL of the reservoir by reducing the water supply. Secondly, the interval between the beginning and end of the pre-flood season is only 54 days, and the main flood season still maintains the FLWL unchanged. Based on the above considerations, only the FLWL of the post-flood season is raised.

**Table 7.** Operation result of each frequency and start counting water level.

| Flood Frequency | Starting Water Level (m) | The Pre-Flood Season | | The Post-Flood Season | |
|---|---|---|---|---|---|
| | | Maximum Storage Capacity (hm³) | Maximum Water Level (m) | Maximum Storage Capacity (hm³) | Maximum Water Level (m) |
| | 185.0 | 993.25 | 186.36 | 980.46 | 186.04 |
| | 185.5 | 1005.77 | 186.66 | 993.20 | 186.36 |
| | 186.0 | 1018.87 | 186.97 | 1005.98 | 186.67 |
| The 1000-year flood | 186.5 | 1032.06 | 187.28 | 1014.80 | 186.88 |
| | 187.0 | 1044.33 | 187.55 | 1032.25 | 187.28 |
| | 187.5 | 1057.36 | 187.84 | 1050.40 | 187.69 |
| | 188.0 | 1075.97 | 188.25 | 1069.04 | 188.1 |
| | 185.0 | 1024.66 | 187.11 | 1010.22 | 186.77 |
| | 185.5 | 1037.53 | 187.4 | 1022.30 | 187.05 |
| | 186.0 | 1049.80 | 187.68 | 1035.69 | 187.36 |
| | 186.5 | 1063.56 | 187.98 | 1049.52 | 187.67 |
| The 10,000-year flood | 187.0 | 1077.86 | 188.29 | 1063.88 | 187.99 |
| | 187.5 | 1092.72 | 188.62 | 1078.80 | 188.32 |
| | 188.0 | 1107.96 | 188.96 | 1094.08 | 188.65 |
| | 188.5 | 1123.33 | 189.3 | 1105.61 | 188.9 |
| | 189.0 | | | 1121.48 | 189.25 |

### 4.3. Risks and Benefits

The paper calculates the FLWL for the post-flood season in 0.5 m increments to obtain probability of exceedance and its increases for different starting water levels (Figure 3), and the expected risk and expected risk increases (Figure 4). When the FLWL is set at 187.5 m in the post-flood season, the maximum probability of exceedance is $2.52 \times 10^{-5}$, which is less than the reservoir calibration flood of $1 \times 10^{-4}$. Compared to the original flood level, the probability of exceedance increases by $2.34 \times 10^{-5}$. From Figure 3, it can be seen that when the FLWL is 185 m~186 m, the probability of exceedance changes very little and the trend of increasing probability of exceedance is moderate. However, when the FLWL is 186–187.5 m, the probability of exceedance tends to increase abruptly. The probability of exceedance of the average FLWL of 186 m in the post-flood season is $0.31 \times 10^{-5}$, which is an increase of $0.13 \times 10^{-5}$ compared to the original level. From Figure 4, it can be seen that when the FLWL is set at 185–186 m, there is little trend in the expected risk of flood protection. However, when the FLWL is set at 186–187.5 m, the expected risk increases steeply.

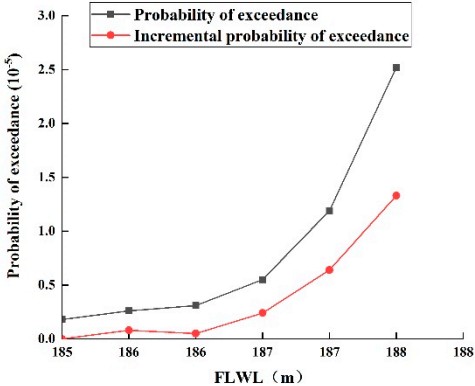

**Figure 3.** Probability of exceedance and increases in the post-flood season.

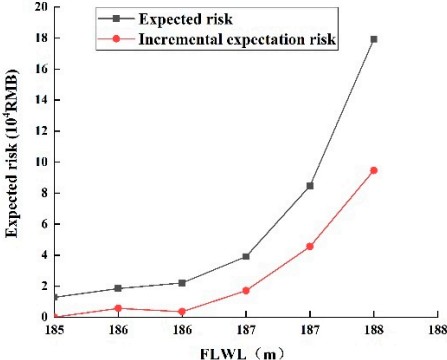

**Figure 4.** Expected risk and expected risk increases at different water levels.

The risk-benefit analysis was carried out by setting the flood level of the Chengbi River Reservoir at 186 m in the post-flood season; the average increase of beneficial capacity is 0.04 billion m$^3$. With the information about water supply and power generation from 2019 to 2020, the increase of beneficial capacity is used for power generation or water supply was calculated. Then, the average annual increase in benefits is between 0.88 and 1.62 million RMB. The increase in expected risk of 0.0093 million RMB relative to the original level when the FLWL during the post-flood season of the reservoir is 186 m. And after taking into account the increased inundation losses of 0.2171 million RMB at that level, the sum of the average annual expected increase in risk is 0.2264 million RMB, which is much less than the expected benefit.

## 5. Discussion and Conclusions

Research on reservoir flood staging and FLWL is significant for improving the utilization of water resources. The study selects multi-year average daily rainfall time series, multi-year maximum rainfall time series, multi-year average daily runoff volume time series, and multi-year maximum daily runoff volume time series as index factors, and uses the fractal method to stage flood season. Finally, the effect of phased dispatching is evaluated in relation to the Benefit-risk theory. The results are as follows:

The pre-flood season of Chengbi River Reservoir is from 13 April to 6 June, the main flood season is from 7 June to 9 September, and the post-flood season is from 10 September to 31 October. Considering the irrigation water demand and flood control risk around the reservoir area, the FLWLs in each stage was finally determined to be 185 m in the pre-flood season, 185 m in the main flood season, and 185~187.5 m in the post-flood season. It is found that when the FLWL in the post-flood season is set at 186 m, the probability of exceedance after reservoir operation by stages in the flood season increases by $0.13 \times 10^{-5}$, the average annual expected risk is 0.2264 million RMB. However, the average annual increase in benefits is 0.88 to 1.62 million RMB.

Compared with the research results of existing scholars [34–36], the present study classifies the flood season to the daily scale with improved accuracy. When the average daily rainfall was used to stage the flooding of the Chengbi River reservoir, the multi-year average series differed from the multi-year maximum series by about five days. Using daily runoff for flood staging, the maximum deviation of the multi-year average series and multi-year maximum series results is about 10 days. The difference between the calculated results of the average daily rainfall time series and the average daily runoff time series is 5 to 10 days. It suggests that the selection of different factor indicators can have an impact on flood staging.

In this study, the staging and scheduling of reservoir floods and the determination of FLWL are investigated. The innovation of the paper is that fractal methods and multiple index factors are used to divide the flood season into daily scales, which improves the staging accuracy. However, there are still some shortcomings in the study. Improvements are needed in the following areas. Firstly, in terms of flood staging, it is recommended that

multiple methods of staging should be used and then mutually validated because of the uncertainty and complexity of hydrology. Secondly, the indirect benefits of tourism and aquaculture due to increased storage capacity have not been calculated because of limited information. Finally, this study is based on long-term rainfall and runoff data, but one direction of research on reservoir scheduling is to forecast the future based on short-term data [37], and how to combine long-term and short-term data needs to be studied further in the future.

**Author Contributions:** Conceptualization, C.M., Z.X. and G.S.; formal analysis, X.L.; funding acquisition, C.M.; methodology, Y.R.; resources, C.Z. and X.L.; writing—original draft, C.M. and C.Z.; writing—review and editing, Y.R., Z.X. and G.S. All authors have read and agreed to the published version of the manuscript.

**Funding:** The authors are grateful for the support of the National Natural Science Foundation of China (51969004, 51979038 and 51569003), the Guangxi Natural Science Foundation of China (2017GXNSFAA198361), and the Innovation Project of Guangxi Graduate Education (YCBZ2019022).

**Institutional Review Board Statement:** Not applicable.

**Informed Consent Statement:** Not applicable.

**Data Availability Statement:** Some or all data, models, or code generated or used during the study are proprietary or confidential in nature and may only be provided with restrictions. We will conduct further research on this aspect in the future. The existing research data will be gradually developed in the subsequent papers, and it is only temporarily confidential at present.

**Conflicts of Interest:** The authors declare no conflict of interest.

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
