# Peer review of "Study on the Staged Operation of a Multi-Purpose Reservoir in Flood Season and Its Effect Evaluation"

_water, doi:10.3390/w13182563_

Round 1

Reviewer 1 Report

Dear authors,

Please, read the attached file containing some suggestions to improve your paper.

Your reviewer 

Author Response

We have carefully read the thoughtful comments from you and found that these suggestions are helpful for us to improve our manuscript. On the basis of the enlightening questions and helpful advices, we have now completed the revision of our manuscript. The itemized responses to your comments please see the attachment. We hope that all these corrections and revisions would be satisfactory.

Reviewer 2 Report

Study on the staged operation of a multi-purpose reservoir in flood season and its effect evaluation

- line 12 - what's mean ”utilisation of flood” (little parenthesis); this is not a common words group for many countries from the temperate climate, and must be detailed; we know very well the flood phenomena represents/ed something vital for many countries, regarding the agriculture (various crops), but for other countries this is not a welcome natural phenomenon; 

- line 13 - the sentence ”For a case study of the Chengbi River reservoir in China.” is not clear (have no verb); 

- line 17 - you cannot start a phrase with ”and” word; probably, you intended to use ”Also”; 

- lines 20-21 - again, a sentence without verb; seems to bi a title :) ;

- line 73 - ”Analytical Methods” in which purpose?; you must complete this sentence;

- lines 77-78 - ”The occurrence of seasonality and timing 77 of floods can be considered to have similar mechanisms ...”, this depends to the studied areas and their climatic characteristics; 

- line 99 - ”2.2.2. Relation Curve Between Water Level and Discharge” ... probably, discharger;

- line 129 - what's mean ”earth-rock”?; probably, a combination between impermeable and hard rocks and soil or non-cohesive rocks?; do you want to say rough rocks?; this means weight dam;

- Figure 1 - China boundary is absent from your map; also, in China, the lower altitudes of relief are represented in blue, not in green?;

- line 199 - what phrase or sentence is ”So that the water level in the reservoir to maintain the FLWL.”?; 

- line 266 - normally, in a paper first, you have the Discussions and to the final Conclusions;

- many calculations and theory, for a little exercise and science; a dry and normal paper; more transparency and interesting elements, probably even more useful to the authorities using the water resource, could have been obtained through a complex analysis of the reservoir dispatcher chart (multi-use) correlated with the water resources and water needs of the territory charts; some thematic maps could have generated attractiveness for the reader; for the local authorities your results can be useful (it depend of their technical preparation), but it is not for a complex and global decision.

Author Response

(The authors gave the same response as above.)

Reviewer 3 Report

This paper addresses deeply important issue, even more in a context of climate change: impact of reservoir flood management on water availability. 

Authors propose a methodology based on a set of indices and the segmentation of flood season to define the reservoir management (through water level definition).

The manuscript is well structured. Cited references are relevant and sufficently contemporary.

Some aspects could be clarify in order to improve the manuscript:

  1. It is not clear the reservoir operation policy applied to evaluate the water level during the main flood season.
  2. It would be useful to explicitly consider the climate change  in the index factor calculation. At least, it could be expressed how the climate change could be incorporated in the index factor determination. Maybe some considerations about stochastic index factors?
  3. The methodology was applied to one case study. Authors could provide some comments about generalization of the methodology.

Author Response

(The authors gave the same response as above.)

Round 2

Reviewer 1 Report

Thank you for taking into account my comments and suggestions. 

No other concerns from my part.